# Parental Burnout: A Progressive Condition Potentially Compromising Family Well-Being—A Narrative Review

**DOI:** 10.3390/healthcare13131603

**Published:** 2025-07-04

**Authors:** Patrik M. Bogdán, Katalin Varga, Lívia Tóth, Kristóf Gróf, Annamária Pakai

**Affiliations:** 1Doctoral School of Health Sciences, Faculty of Health Sciences, University of Pécs, 7621 Pécs, Hungary; livia.toth@etk.pte.hu (L.T.); grof08@gmail.com (K.G.); 2Department of Affective Psychology, Faculty of Education and Psychology, Eötvös Loránd University, 1064 Budapest, Hungary; varga.katalin@ppk.elte.hu; 3Institute of Emergency Care, Pedagogy of Health and Nursing Sciences, Faculty of Health Sciences, University of Pécs, 7621 Pécs, Hungary; annamaria.pakai@etk.pte.hu

**Keywords:** family, burnout, parental burnout, parental stress, exhaustion, anxiety

## Abstract

Background: Parental burnout is one of today’s significant challenges, increasingly manifesting as a problem in our fast-paced world. The aim of this review is to create an exploratory, descriptive summary of parental burnout through the analysis of available international publications, providing a clearer and more accurate understanding of the psychological condition, severity, manifestations, and treatment options. Methods: Our narrative literature review includes publications from 2010 onwards, focusing on those that directly address the topic of parental burnout syndrome and contain epidemiological data, risk factors, symptoms, diagnostic possibilities, and treatment strategies. We excluded publications that examined the condition within narrow societal groups, such as parents caring for children with somatic mental disorders. Results: Based on our review, it appears that parental burnout may potentially affect both women and men. Factors such as low emotional intelligence, workplace stress, and lack of supportive family background render parents vulnerable to this condition. Significant differences in the prevalence of parental burnout can be measured between countries, due to cultural differences. Parental burnout has extremely detrimental effects on family dynamics and the emotional development of children, and it can negatively impact the willingness to have more children at the family level, which has dire consequences considering the low birth rates characteristic of European countries.

“Family is not an important thing…It’s everything.”(Michael J. Fox)

## 1. Introduction

The World Health Organization (WHO), upon its establishment in 1948, defined health as a holistic entity characterized by complete physical, mental, and social well-being, rather than merely the absence of disease [1]. In today’s fast-paced and stimulus-filled world (caused by the digital environment, social media, information overload, etc.), numerous factors can negatively impact our daily lives and mental health [2]. The pressure to meet expectations can be a constant source of stress not only in the workplace but also within the family [3]. This is because the parental role demands continuous compliance, availability, and high-quality care for the child(ren). However, children may also affect parents through daily activities, behavioral and health issues, and later, educational challenges [4]. The concept of parental burnout (PB) refers to a syndrome affecting both men and women, characterized by prolonged physical and mental strain and exhaustion associated with the parental role, which may be accompanied by emotional detachment from the child, overwhelming exhaustion, and even self-doubt regarding one’s suitability in caregiving and nurturing roles [5]. Although the *DSM-5* (*Diagnostic and Statistical Manual of Mental Disorders, Fifth Edition*) does not address burnout as a separate diagnostic category, nor parental burnout specifically, the World Health Organization generally recognizes burnout as a phenomenon related to work in the *ICD-11* (*International Classification of Diseases, 11th Revision*), under the category “factors influencing health status or contact with health services [6,7].

The ability to cope with everyday stress varies among individuals, but scientific research has shown that greater coping skills positively impact mental health [8]. It is now well established that societal expectations for parenthood, the pressure to meet the child’s needs, workload, and the resulting decrease in energy levels are all risk factors for parental burnout [9]. Additionally, the lack of external support, single parenthood, a parent’s chronic illness, the number of children, being too young or too old as a parent, excessive workplace stress, previous or existing mental health issues (such as depression or generalized anxiety), or caring for a child with a chronic condition can all increase vulnerability to this condition [10]. In parents raising children with somatic or mental illnesses, the level of “PB” can be much more severe, as research findings indicate that the postnatally recognized genetic disorders cause even greater stress levels for the parents and jeopardize the early bonding between mother and child [11,12]. Parental burnout is influenced by numerous sociocultural and economic factors, which increasingly challenge and burden the experience of parenthood [13]. In the long term, psychological symptoms may manifest as physical symptoms, such as pathological fatigue, sleep and attention disorders, or even somatic pain. In severe cases, suicidal thoughts may develop.

However, parental burnout affects not only the parents but also the child(ren). Every child has the right to a loving family environment, but a parent suffering from burnout may be able to provide this environment only to a limited or inadequate extent. This can lead to attachment problems with the parent in early childhood, which may compromise the child’s emotional development and are closely related to it, thereby influencing the child’s psychological, cognitive, and social development [14]. As a consequence of “PB,” children may experience severe behaviors that significantly harm their emotional and mental development, such as neglect and physical abuse. Children raised in such an environment may develop post-traumatic symptoms, and later in life, they may display harmful behaviors such as self-harm, addictions, and violent behavior in adulthood [15]. Although data on the lifetime prevalence of the condition is scarce, European studies estimate the prevalence to be around 5% [16]. While this figure may seem modest, the problem is exacerbated by the limited number of publications available from Hungary. Hamvai and colleagues estimated the prevalence in the Hungarian population to be between 2 and 12% [17].

The aim of this review is to create an exploratory, descriptive summary of parental burnout through the analysis of available international publications, providing a clearer understanding of the disease’s characteristics, severity, and manifestations. Additionally, we aim to investigate and detail the diagnostic options, risk factors, and how these factors impact parents’ mental health. We seek to explore the supportive needs of parents and formulate useful and constructive recommendations based on the reviewed literature, which could be integrated into primary prevention efforts. We will also attempt to summarize the treatment options currently available for those affected by the condition.

## 2. Materials and Methods

### 2.1. Inclusion and Exclusion Criteria

Since this paper is a narrative analysis of the literature, we collected data from existing and available publications and analyzed them. We did not conduct any independent data collection or analysis. In our research, we reviewed and analyzed publications from 2010 onwards. The reason for this is that these studies accurately reflect the current social and cultural environment, making these publications a more credible representation of the effects that parents are experiencing in today’s society. Naturally, older publications were also processed and referenced to frame the examined phenomenon theoretically. The identified literature was published in peer-reviewed journals, ensuring that we met the quality standards expected of scientific works. We selected literature that directly addresses parental burnout syndrome and includes epidemiological data, risk factors, symptoms, diagnostic possibilities, and treatment strategies. Additionally, we considered it important to analyze articles that explore the relationships between the disease and sociodemographic or sociocultural variables. We excluded publications that examined the condition within narrow societal groups, such as parents caring for children with somatic or mental disorders (e.g., Down syndrome, mental disability, diabetes mellitus, cystic fibrosis). The reason for their exclusion was that, for example, parents of children with somatic or mental illnesses may experience extreme and continuous stress. Additionally, the causes and mechanisms of burnout in their case may differ from those in the general population. Therefore, the results of studies conducted within this group would not necessarily be comparable, and as a result, we would not be able to reliably determine the general characteristics of burnout.

### 2.2. Research Strategy

We collected the relevant literature from officially available online databases such as PubMed, Google Scholar, LAM, EBSCO, Wiley Open Library, MDPI, Cochrane Library, SpringerLink, and ScienceDirect. To ensure the broadest possible source collection, we conducted the literature search using a large number of keywords and logical operators.

After precisely defining the aims of the research, we considered the publication year and scientific credibility of the sources for selection. Following a review of the abstracts, we listed the articles that appeared suitable for our study. After a complete review of the literature, we selected those publications that we deemed the most credible in terms of methodology, results, and conclusions. Our criteria focused on an adequate sample size, the use of validated and standardized measurement tools, and the application of appropriately chosen statistical methods. Out of approximately 130 initially collected sources, 47 were ultimately included in our analysis.

## 3. Results

### 3.1. The Epidemiology of Parental Burnout

Based on the summarized findings of scientific research, at least 5% of families in English-speaking countries suffer from parental burnout [18]. In Eastern countries, there are limited data available regarding the prevalence of this condition. Furthermore, as parental norms, societal expectations, and childcare practices vary across different countries, it is likely that the level of burnout in a given population also varies worldwide [19]. Another challenge in obtaining accurate estimates arises from the fact that different studies have established varying cutoff scores for the tools used to identify the condition and assess its severity. Consequently, prevalence rates can range from as low as 1% to as high as 30% in some studies. It is worth noting that the variability in occurrence is also explained by the fact that many studies target the general population, while others focus specifically on parents of children with chronic illnesses [20]. In 2021, the Belgian psychologist Isabelle Roskam and colleagues conducted a study encompassing 42 countries, with a sample size of 17,409 participants (12,364 women and 5045 men), to gain a broader perspective on the prevalence of parental burnout. As this remains the largest study on the subject to date, it should be referenced as the gold standard for our research. Their findings indicated that the prevalence of the condition in the United States was 8.9%. In European countries, the prevalence was 9.6% in Poland, 1.8% in Germany, 1.6% in Austria, 2.6% in Sweden, 7.1% in Switzerland, and 6.2% in France. The highest prevalence was recorded in Belgium, at 9.8%, while Italy had the lowest, at 0.6%. In Middle-Eastern countries, the prevalence was 0.4% in Turkey, 5.5% in Lebanon, and 1.5% in Iran, with no measurable cases in Pakistan. In East Asia, the prevalence rates were 1.4% in China, 2.8% in Japan, 0.7% in Vietnam, and the lowest rate was recorded in Thailand, at 0.2% [21].

### 3.2. Assessment of Parental Burnout

An expert-developed questionnaire is available for the detection of parental burnout. This test helps determine whether an individual is affected by the condition, and it can provide information about the severity of their burnout. The questionnaire is called the “Parental Burnout Assessment” (PBA), developed by the Belgian researcher Isabelle Roskam and her colleagues in 2018. The PBA consists of 23 questions that are organized into three subscales: emotional exhaustion, emotional distancing, and feelings of failure in the parental role. These dimensions were created using a deductive approach, based on Maslach’s three-dimensional structure of workplace burnout [22]. Each of the 23 questions is rated on a 0–7-point Likert scale, reflecting how often the respondent experiences or thinks about the statements presented. The tool does not include any reverse-coded items. Since its release, the questionnaire has been translated into multiple languages, and all adaptations have demonstrated high measurement reliability. The PBA plays a crucial role in the early detection of parental burnout, making it essential to implement its use more widely in settings such as general practitioners’ offices, maternal and child health services, and psychologists’ practices.

### 3.3. General Risk Factors for Parental Burnout

To understand the risk factors that contribute to and exacerbate parental burnout, it is essential to provide an appropriate theoretical framework for the phenomenon. Without a solid theoretical foundation, we cannot comprehend how these risk factors induce negative changes in the physical and mental equilibrium of health. The theoretical background of parental burnout is closely related to the “demand–resource relationship” model previously applied in organizational psychology to conceptualize workplace burnout. According to this model, burnout occurs when workplace demands are high but resources are low [23]. In the case of parental burnout, however, it is more precise to state that this condition arises when parental resources are insufficient to meet any demands, regardless of their nature. In a publication by Mikolajczak and Roskam in 2018, the authors discuss several stress-inducing factors, categorized as demands or risk factors, including parental perfectionism, an excessive amount of parental responsibilities, an overload of household duties, poor parenting practices, lack of social support, and low emotional intelligence. Conversely, they identified protective factors such as high intelligence, strong empathy, an optimal work–life balance, and the presence of external support, whether from a family member or a family friend [24].

### 3.4. The Role of Sociocultural Factors in Parental Burnout

Sociocultural factors refer to the set of social and cultural characteristics that significantly influence an individual’s thinking, lifestyle, value system, decisions, and perception of the world they live in. For this reason, these factors play an important role in the development of parental burnout or in shaping its characteristics.

#### 3.4.1. Education Level

Educational attainment determines a parent’s or family’s socioeconomic status; as such, these two factors are closely intertwined. A parent with a higher level of education may approach their parental role from a broader intellectual perspective, which can enhance their mental coping mechanisms in the face of stress. In contrast, lower educational attainment is generally associated with lower income, and families living under poor financial conditions may be exposed to higher levels of stress, as securing their basic daily needs can pose a serious challenge alongside the difficulties of raising children [25].

#### 3.4.2. Social Support

Social support is considered to be one of the most important influencing factors in the development of parental burnout. This is because humans are inherently social beings, with a fundamental need for emotional and intellectual connection with others. In experiencing and coping with stress, it is crucial whether the mental or physical burden can be shared between two people—typically a woman and a man. Multi-generational families have an even greater advantage in this regard, as grandparents can also take on supportive roles in managing child-rearing responsibilities. Social support may also extend beyond the immediate family to the level of social capital, encompassing resources from friendships or community-based networks (such as support groups) that manifest in assistance provided to the family—regardless of the form that this support takes [26].

#### 3.4.3. Parenting Styles

There is also a strong correlation between parenting attitudes—such as permissive, democratic, and authoritarian parenting styles—and the level of parental burnout. In parenting approaches that are commanding, based on instructions, and less responsive to the emotional needs of the child(ren), conflicts between parent and child are more common. These conflicts negatively influence how stress is experienced and processed. In contrast, when parenting is grounded in mutual respect, democratic decision-making, and consistent rules, and when the parent–child relationship is characterized by emotional warmth, it can enhance the parent’s self-image and increase self-satisfaction. Applying such a model may lead to a more balanced mental state for the parent and also ensures a stable and nurturing family environment for the child(ren) [27].

### 3.5. Gender Differences in Parental Burnout

In a study conducted by De Santis and colleagues in Brazil in 2024, the aim was to examine the differences in parental burnout between mothers and fathers. The authors hypothesized that female participants would be more vulnerable to the condition, attributing this to the greater energy women that invest in parenting activities. Their findings confirmed that the women in their sample scored significantly higher on burnout measurement questionnaires. Interestingly, even in families where paternal involvement was equal to maternal involvement, the incidence of burnout among women was still higher. The researchers suggested that this could be due to significant differences in stress-coping abilities between genders, as well as women’s greater susceptibility to emotional regulation disorders, leading to poorer management of stress related to parenting [28]. In today’s society, household tasks are now shared between men and women in a significant portion of households. However, the dominance of mental labor still persists, meaning that women continue to be primarily responsible for family decision-making, problem-solving, memory, planning, task coordination, and timely completion of tasks [29]. Another concerning issue is that, according to WHO surveys, at least one-third of women living within families have experienced domestic violence, which can manifest in physical, emotional–mental, sexual, and verbal forms. Domestic violence is associated with numerous harmful mental health issues, ranging from depression and anxiety to substance abuse and even self-harm. This can also contribute to women being more vulnerable to parental burnout, especially if violence occurs within the family or even becomes a systemic pattern [30]. A study conducted in France, involving 900 participants, yielded results similar to those of De Santis’s research. However, an interesting distinction emerged in that women only became affected by burnout after the “demands” (risk factors) exceeded their maternal resources, whereas fathers exhibited signs of burnout even before their resources were fully depleted. A critical observation from this research is that, despite the differences between genders, the symptoms of the condition were identical for both mothers and fathers. Therefore, it can be concluded that parental burnout does not exist as a separate phenomenon for mothers or fathers, but rather as a general parental issue [31].

### 3.6. Cultural Differences in Parental Burnout

Culture is defined as a community where shared ideas, belief systems, values, and customary practices exist [32]. Cultures vary by geographical region and are influenced by factors such as ethnicity, religion, and urban or rural living conditions [33]. Both physical and mental health are supported by various cultural conditions, with family and a nurturing family atmosphere serving as fundamental building blocks [34]. It can also be stated that the effectiveness of individual health promotion depends greatly on the social and cultural characteristics of families [35]. Thus, it is understandable that families affected by parental burnout experience the condition, its severity, and the capacity for rehabilitation differently based on their culture. Research findings indicate that the cultural variance in parental burnout primarily depends on whether a country has an individualistic or collectivistic social structure. The former is characteristic of the United States, Canada, Australia, the United Kingdom, and Western European countries, while the latter is typical of Indonesia, China, Japan, Korea, and India. However, due to intense urbanization, it is possible that individualistic traits may become more prominent even in a fundamentally collectivist country. The main difference between these two perspectives is how individuals view themselves [36]. In collectivistic societies, individuals and other group members form an inseparable unit, dependent on one another, with community goals or problems taking precedence over individual goals or issues [37]. In contrast, in individualistic societies, the individual is seen as a separate, independent entity from the community, where community problems are often irrelevant from the individual’s perspective [38].

In a 2023 study by Matias and colleagues, the extent of parental burnout and its variation across cultural differences were mapped in 36 countries. A significant portion of participating parents fell into the low-burnout category of “fulfilled parents,” primarily consisting of men, working parents with a single child, or those living in multi-generational families. Sociodemographic factors and the level of burnout showed the least variance in the sample. However, in individualistic countries, there were significant differences in the feeling of losing the parental role; in other words, they felt that they had failed as parents. Emotional exhaustion was also found to be higher in these countries, reinforcing the assertion that societies in individualistic countries are more vulnerable to parental burnout. While the study hypothesized certain predictors that may play a dedicated role in the development of burnout, no such distinct predictors were identified between the two cultural groups [39]. Of course, cultural differences can also vary considerably between countries or regions in terms of the value and esteem in which the role of the family is held in a given society, the social customs of families, the emphasis placed on traditions, and religious and spiritual differences. All of these factors play a role in the perception of stress and in its successful or unsuccessful coping.

### 3.7. Stages of Parental Burnout

Parental burnout is a response to prolonged and chronic stress experienced by parents [40]. The concept of stress experienced by parents was first described and theoretically framed by Abidin in 1992. He defined parenting stress as the burden caused by the characteristics of both the parent and the child, influenced by personality, attachment patterns, underlying health conditions, and adaptability. These factors determine the extent of the burden that parents feel in their parenting role [41]. Deater-Deckard described this as an adaptation process, where parents attempt to adjust to the demands of parenting but develop aversive psychological and physical reactions during this process [42]. While parental burnout and parental stress are closely related, they should not be regarded as identical phenomena [43]. Various emotional and somatic responses to prolonged stress can cluster into symptom groups related to burnout [44]. The three primary symptom groups include emotional exhaustion, emotional distancing from the child, and feelings of failure in the parental role [45]. As with the progression of any illness, the development of parental burnout should be understood as a process in which the condition and its symptoms worsen over time. In McEwen’s theory of chronic stress, exhaustion is a primary response, suggesting that the first stage of parental burnout also manifests as exhaustion [46].

In a study conducted by Roskam and Mikolajczak in 2021, the researchers aimed to map and examine the nature of the process of parental burnout’s development through a three-wave longitudinal study. Despite extensive literature research, this study was the only one found here that examined the long-term evolution of the variables associated with parental burnout. Their findings suggested that parental burnout is not an instantaneous occurrence but, rather, the result of a prolonged process. They identified emotional exhaustion as the initial stage.

The first stage is characterized by the parent who is constantly tired, even when getting out of bed in the morning, realizing that they must spend another day with their child(ren). This is accompanied by a kind of emotional depression and a feeling of being drained. The parent may feel that they have reached the limit of what they can give to their role as a parent. This evolves into a second phase called emotional distancing, in which the relationship between parent and child changes in a negative direction, with the parent becoming increasingly emotionally distant from their child(ren). They involve their child(ren) less and less in their daily lives, and even the contact between them is limited to meeting the child’s minimal needs and basic requirements. The third and most serious phase is the loss of accomplishment in one’s parental role. This phase is characterized by the parent feeling that they have had enough of parenthood, no longer finding joy in their child(ren), and being unable to continue in their parental role. As a practical conclusion, the authors emphasized that, due to the processual nature of the condition, special attention should be given to fatigued parents in their parenting activities. This highlights the significant role of general practitioners, midwives, and psychologists [47].

### 3.8. Consequences of Parental Burnout

A nurturing and loving family environment is essential for a child’s emotional development and mental health [48]. Additionally, the close emotional bond between the child and the caregiver is of paramount importance [49]. One of the most severe negative consequences of parental burnout is the development of emotional distancing between the parent and child [50]. A barren or emotionally impoverished environment has been shown to negatively affect the parent–child relationship. Gillis and Roskam conducted a longitudinal study in 2019 to examine how parental exhaustion affects the parent–child relationship for both mothers and fathers. They also investigated how the partner role influences this phenomenon. Their findings indicated that the emergence of exhaustion led to an immediate and significant deterioration in the parent–child relationship, although the level of exhaustion did not necessarily reach a critical point. For mothers, partner support was able to mitigate negative emotional effects, but only as long as the exhaustion was low [51]. Another significant consequence of parental burnout can be neglectful or even violent behavior from parents. Parental violence can range from minor to major physical aggression or psychological aggression [52]. Neglect is defined as an inappropriate or even rejecting response to a child’s needs, which also harms the child’s social and emotional development [53].

In a study conducted by Hansotte and colleagues in 2020 [54], parents were categorized into three profiles based on emotional distancing, emotional exhaustion, and loss of effectiveness. They investigated whether the level of parental burnout in these profiles was correlated with neglectful or violent behavior. Their findings revealed that in parental profiles where significant levels of burnout were present, neglect and all forms of violence occurred more frequently, with the exception of physical abuse. The authors suggested that a conscious and highly useful inhibitory factor might be at play, which, while not preventing parents from committing verbal or psychological aggression against their children, nonetheless prevents physical abuse. This could be attributed to increasing stigmatization and legal repercussions associated with physical violence [54].

Another pressing issue is the declining fertility rate, which is characteristic of Western societies and is a general concern today, and one of its explanations and causes could be parental burnout [55]. The willingness to have children is influenced by various factors, including age, family financial situation, economic environment, social background, and parental support [56]. For parents who already have children, the desire to have more children is significantly negatively impacted if the child requires special care [57]. In a 2021 publication, Piotrowski explored the notion that some parents may even regret parenthood. The degree of this regret showed a close correlation with the severity of parental burnout and was associated with unexpected childhood events, poorer mental or physical health, or whether the parent was experiencing a crisis of parental identity [58]. While it is difficult, if not impossible, to rank the severity of the consequences of parental burnout, one of the worst and potentially most harmful outcomes is the emergence of thoughts related to “escape,” which in this case refers to leaving the family or, in the worst-case scenario, contemplating suicide [59].

### 3.9. Prevention and Treatment of Parental Burnout

If parental burnout is considered to be a mental health issue, it is essential to prevent its onset, just like any other illness. This means that primary prevention is crucial [60]. The first step is to mentally prepare parents for parenthood, as couples who choose to have children will face entirely new challenges [61]. In today’s idealized world, discussions about parenthood often emphasize only the positives, while the difficulties and challenges of parenting receive little attention [62]. Therefore, it is vital for couples anticipating parenthood to have a complete understanding that being a parent sometimes involves significant struggles, sacrifices, and daily emotional coping, which can greatly impact parental functions and the emotional development of the child(ren) [63]. One of the simplest ways to provide this information is for the parents of couples expecting a child to share their experiences openly. Friends and acquaintances with children should also engage in honest conversations. The role of the midwife is critical, as expectant mothers visit the midwifery practice multiple times during pregnancy, making the midwife an essential player in mental preparation. This, of course, applies only to those countries where this system is an integral part of the healthcare infrastructure.

It can also be beneficial for couples to seek the help of a psychologist before becoming parents. Even before starting a family, it would be important to emphasize—and ideally, to teach within the framework of developmental courses—the adoption of appropriate parenting attitudes, with a particular focus on mastering the methods of authoritative parenting. Special attention should be given to single parents lacking social support, as well as to families with low income and low educational attainment, since these factors all represent risk elements in both the development and the more severe manifestation of parental burnout. For those already experiencing burnout, mindfulness techniques can be beneficial [64]. Parenting can cause stress from many angles, but being present in the moment and non-judgmentally managing negative thoughts, consciousness, and emotions can serve as effective stress-reduction methods [65].

Cognitive behavioral therapy, which is effective in reducing anxiety and depression, can also be successfully integrated into the treatment of parental burnout [66]. A core principle of cognitive behavioral therapy (CBT) is that, when negative events occur, we respond not directly to the events themselves but to the interpretations and meanings associated with them. CBT can assist us in recognizing, challenging, and modifying unhelpful thoughts, and it can also be used to acquire behavioral strategies that promote more effective and adaptive responses [67]. Cognitive behavioral therapy can help individuals reinterpret the meanings associated with negative events experienced during parenting, thereby reducing the intensity of harmful emotional reactions caused by the recurrence of those events [68]. Participating in relaxation training, meditation, and yoga therapy can also be effective in preventing or treating burnout [69]. The ability to adapt to stress can be significantly enhanced through various coping strategies [70]. In terms of parental burnout, strengthening problem-focused coping is the goal, where parents direct their thoughts and efforts towards recognizing the problem, its source, and possible solutions. Effective problem-focused coping has been shown to positively influence burnout [71]. Another significant protective factor and potential stress reducer for parents is ensuring that they have adequate time and quality for leisure activities, allowing them to mentally rejuvenate and relax. This naturally requires appropriate family support, particularly from grandparents, who can provide supervision for the child while the parents are away. For parental couples, emotional support for one another can serve as a strong bulwark against everyday stressors.

### 3.10. The Relationship Among Parental Burnout, Depression, and Postpartum Depression

Depression, postpartum depression, and parental burnout are interconnected through their core symptoms, as all three are associated with low emotionality, depression, exhaustion, lack of motivation, and even attachment disorders in the relationship between mother and child. Special attention should be given to parents who have experienced a depressive episode at any point in their lives, as even after recovery from depression, they become more vulnerable to relapse. Considering the increased mental load and demands of parenthood, the risk of relapse must be taken very seriously in all aspects. An additional problem arises if a parent struggling with depression has a child, which can significantly exacerbate their symptoms, even during pregnancy [72]. Postpartum depression manifests after the tenth day following childbirth and typically lasts about 3 to 6 months, presenting as a depressive syndrome. One severe consequence of PPD is that it can cause attachment disorders between the mother and her child, with long-term negative effects. The strength and quality of the mother’s attachment play a crucial role in meeting the child’s needs, serving as a resource for responding to these needs. In this way, postpartum depression can predict the development of parental burnout, and it may even manifest earlier, forming a mutually reinforcing cycle that damages the mother’s mental health. Since postpartum depression increases the risk of burnout, which, in turn, exacerbates the severity of depression, these conditions can perpetuate and intensify each other [73].

Distinguishing the illnesses discussed above from parental burnout is the task of clinical professionals; however, there are certain differences that define clear boundaries. For example, in major depression, mood disturbance is generally a characteristic feature, including a persistent and long-lasting feeling of sadness and decreased energy levels, which are not specifically related to the parental role but occur broadly. Postpartum depression is distinguished by the fact that, according to the literature, it typically resolves after six months, whereas the symptoms of parental burnout may either appear afterward or may develop from postpartum depression, with the symptoms persisting thereafter.

## 4. Discussion

The aim of this paper was to identify and summarize the causes, risk factors, consequences, and potential methods for the treatment or prevention of parental burnout. Our review concentrated not solely on local attributes but also on global-scale determinants and the factors that define “PB”. Based on our findings, it can be concluded that “PB” is a multidimensional phenomenon that can develop as a result of numerous indirect and direct factors [74]. The appearance of symptoms and the severity of individual symptoms show a dynamic progression over time, highlighting the processual nature of burnout, rather than it being a static, acute phenomenon. This underscores the importance of early detection, as identifying the condition at its mildest stage can help halt its progression. The phenomenon of parental burnout can be understood through the well-researched theoretical background of workplace burnout, as both share similar foundations. However, significant differences exist, necessitating the discussion of parental burnout as an independent phenomenon [75]. For instance, while individuals experiencing workplace burnout can often leave a toxic work environment without irreversible consequences, parental burnout does not offer such an exit without severe repercussions for the family and children. Epidemiologically, parental burnout is a common issue, although there are countries where this phenomenon is not measurable. A significant difference can be observed between the data measured in Western and Eastern societies. This pronounced disparity can be explained by the multi-generational family model more commonly seen in Eastern countries, where multiple generations live together. When responsibilities related to children are shared among several people, the burden per individual naturally decreases. In Eastern cultures, family cohesion, shared goals and values, and collective identity are typically more prominent. Child-rearing in Eastern cultures creates a strong bond with family and social values, and the fact that spirituality and religion are more intense there may also explain the lower levels of burnout, as practices associated with religion (e.g., prayer, weekly day of rest) can have stress-relieving effects. A key promoting factor in the development of parental burnout is the discrepancy between the demands posed by parenting tasks and the resources available to meet those demands [76]. While we may not expect significant changes on the demand side, it is anticipated that the burdens on parents’ resources will become increasingly drastic. This decline in resources is partially due to a growing number of parents working multiple jobs or family members taking on second jobs to achieve financial security, negatively impacting the parents’ energy reserves. Among the individual- and family-level risk factors, the most important include a lack of social support from family members, excessive perfectionism, being female, low emotional intelligence, and a predisposition to mental health issues. A history of depressive or anxiety disorders can further exacerbate the situation [77]. The decline in grandparental support, which serves as a preventive factor, is also noteworthy. Many grandparents are working actively or live far from their families, limiting their ability to provide support. Unfortunately, it is increasingly common for grandparents to decline the opportunity to care for their grandchildren due to their own commitments or obligations. It is worth mentioning the “sandwich generation,” which refers to parents who simultaneously care for their children and look after their elderly parents. This increased physical and mental burden can contribute to the development of burnout, especially in cases where the aging grandparents in need of care suffer from illnesses that significantly impair their health. An important point related to this is that having children, especially in families with multiple children, provides the advantage of being able to care for one’s parents when they become elderly. Additionally, with more children, the potential caregiving burden is shared proportionally, helping to ease the overall load. In the parent–child relationship, it is important to recognize that the duties and responsibilities are not solely the parents’ towards the children. The best approach is to establish a living arrangement where the child also fulfills their responsibilities within the family, to the extent that their age and mental/physical abilities allow. This not only can enhance the child’s sense of importance and usefulness but also has the potential to alleviate a significant burden from the parents. Parental burnout is a significant issue, because every child deserves a loving and nurturing family environment. Emotional distancing, abuse, or neglect can have devastating effects on a child’s emotional and psychological development [78]. Moreover, it is essential to discuss the harmful effects on the parents themselves, as burnout negatively influences social relationships, potentially leading to marital breakdowns. In Europe, the number of marriages is stagnating, and this statistic is worsened by the dissolution of marriages attributed to burnout. An additional, yet less extensively examined, factor is the presence of disagreement between partners regarding the desired number of children, which can generate interpersonal tension. Such discord has the potential to contribute to the dissolution of the partnership; however, if a child is nonetheless conceived and born despite this disagreement, the parent who did not wish to have the child may be at increased risk of experiencing parental burnout. Thoughts of “escape” can manifest at various levels of severity, ranging from a desire to leave to contemplating suicide [79]. Special attention should be given to male family members, as men often struggle to open up and share their feelings or thoughts, viewing vulnerability as a weakness or failing to recognize their need for help. While women attempt suicide more frequently, men lead the statistics on completed suicides worldwide [80]. In treating parental burnout, psychologists play a crucial role; however, this requires that the condition be recognized first. Primary care providers, including family doctors and advanced practice nurses, play a significant role in monitoring and screening for signs of parental burnout among families.

Our review highlights that parental burnout requires an interdisciplinary approach, where psychologists, family doctors, social workers, and healthcare professionals work together in a synthesis with the parent or parents. Although our literature review did not primarily focus on parents raising children with somatic or mental illnesses, we feel that it is our duty to emphasize the importance of strengthening support and assistance within this group, as their risk of burnout is higher. We want to stress the significance of social networks and mental health services, as well as the need to make these accessible to all parents. Even raising awareness that a parent feels they need help and knows where to turn can be an effective step forward. We believe that it is crucial to raise public awareness about parental burnout and to draw parents’ attention to its existence—potentially through campaigns—so that they understand that the phenomenon exists and that anyone can fall victim to this psychological state, thus facilitating their seeking help. This research area is almost exclusively based on cross-sectional studies, which make it difficult to examine the dynamics of PB; therefore, we consider conducting longitudinal studies to be important. Given that publications on this topic are extremely limited in Hungary, it is essential that exploratory studies on this subject be initiated in the near future.

## 5. Conclusions

Parental burnout is a significant issue in today’s society. It is anticipated that the prevalence of this condition will rise in the future, largely due to the fact that, in our increasingly fast-paced world, parents are facing a decrease in the resources that they can allocate to their children. This form of burnout has extremely detrimental effects on family dynamics and the emotional development of children. At the family level, it can negatively impact the willingness to have more children, which is particularly concerning given the already low birth rates characteristic of European countries. Early screening and monitoring of families should become a prominent task within the healthcare system, emphasizing the necessity of holistic and team-based collaboration. Addressing parental burnout effectively requires a comprehensive approach that involves multiple stakeholders to support families in navigating these challenges.

## Data Availability

No new data were created or analyzed in this study. Data sharing is not applicable to this article.

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
