# Peer review of "Parental Burnout: A Progressive Condition Potentially Compromising Family Well-Being—A Narrative Review"

_healthcare, 2025, doi:10.3390/healthcare13131603_

Round 1

Reviewer 1 Report

Comments and Suggestions for Authors

Dear authors, I recently had the opportunity to review your manuscript "Parental Burnout: A Slow Poison That Can Crush Even The Most Beautiful Family Idyll: A Narrative Review".

A very timely ad important topic, however I feel your manuscript coudl benefit from incorporating the following suggestions:

Introduction:

  • I suggest you introduce the definition of parental burnout (ll. 53-59) earlier in the introduction, preferably before mentioning influencing factors for parental burnout. The definition could be presented from ll. 45 ff., then describe influencing factors followed by prevalence estimates. The definition should be clear and precise.
  • the introduction would also benefit from a stronger theoretical framework, e.g, a family relationship perspective: why is parental burnout relevant not only for the individual but also other family members, especially children of affected parents? I feel you should go into more detail at this point as the condition specifically emerges in the parenting context

Methods: 

  • please explain why you excluded "narrow societal groups". I suppose you were interested in parental burnout in the general population but this must be better explained and should also be part of your theoretical background in the introduction.
  • please explain why you only analysed studies from 2010 onwards.
  • ll. 95-99: Statements like these are not be part of the methods section. You could transfer it to the limitations and strengths section of the discussion.

Results:

  • ll. 149-152: Another explanation is that while household tasks/ domestic labour is increasingly split between mothers and fathers, the mental labor in mothers is substantially higher than in fathers (e.g., Reich-Stiebert, N., Froehlich, L., & Voltmer, J. B. (2023). Gendered Mental Labor: A Systematic Literature Review on the Cognitive Dimension of Unpaid Work Within the Household and Childcare. Sex roles88(11-12), 475–494. https://doi.org/10.1007/s11199-023-01362-0). Please take this into consideration when interpreting your results.
  • l. 193: Abidin refers to parenting stress, not parental stress. Please change the wording
  • l. 249 f.: Please rephrase, the sentence does not uphold to scientific appropriate language standards. Delete "a phenomenon that exists"

Author Response

Dear Reviewer,  
Please find our detailed responses in the attached file.  
Thank you,  
The Authors

Reviewer 2 Report

Comments and Suggestions for Authors

Title: describing a health condition as a ‘poison’ and using terminology such as ‘crush’ and ‘beautiful’ seems emotional. This language is not usually used in a clinical context (which is the main subject of the journal). Although the title is seemingly captivating and appealing, narration of finding should be made on scientific basis with less emphasis on terms with subjective nature.

Abstract

Line 14: please change the term ‘study’ to ‘review. The same change is advised to be made in line 64 of the background.

Background

Line 47: can the authors explain or provide a specific ‘case definition’ for ‘parental burnout’? I understand the magnitude of variability of classifying a parent as being a subject of burnout. However, making an effort to define or provide a classification for this condition is necessary for scientific clarity purposes.

Methods

Exclamation mark in line 99 may not be suitable for an academic writing. Please revise.

It is not clear how figure one is describing the epidemiology of parental burnout. It is showing literature search process. Please revise.

We also advise the authors to delete figure 1 as it is actually not presenting any important information to the reader and is already described clearly within the text.  

Results

Line 104: what do you mean by ‘conservative estimates’ please elaborate.

The statement made about the prevalence of burnout among parents obtained from Anglo-Saxon studies was mentioned twice ( one in the background and one in the results). This increases the redundancy of the review with unnecessary repletion which should be avoided. Also it is confusing that the same statement was made from two references  ( 10 and 12) .

The authors are advised to make an effort to explain the marked variability of burnout prevalence between European and Middle Eastern countries.

Can the authors provide examples of countries with individualistic or collectivistic social structure? Can a single country have both societal structures? Is it possible to have other cultural concepts that goes beyond these two social structures? For example, a parent who has an individualistic characteristics in a community which is described as a collectivist societal structure?

The section describing parental burnout stages is not clear. By reading this section, the number of stages was not clear, definition of each stage was vague, and only one stage was namely described as first stage without clearly indicating what happens next. This section should either be written to describe clear stages or simply merged with a section describing symptoms of the condition.

Section describing the assessment and diagnosis of the parental burnout should be illustrated earlier in the paper. Additionally, the section is not discussing diagnosis. A questionnaire, even with high validity and reliability is usually used as a screening tool which should be followed up with a clinical diagnosis via an expert in the field.

Section describing treatment of parental burnout can be labelled as (prevention and treatment of parental burnout)

The discussion seems repetitive and does not provide specific contribution to the manuscript. Better organization of the discussion may enhance its coherence and reduce repetition.

Author Response

(The authors gave the same response as above.)

Reviewer 3 Report

Comments and Suggestions for Authors

Thank you for the opportunity to review this manuscript. The topic is important and relevant, but several areas need clarification and improvement:

  1. Definition and Classification: Please include a clear definition of parental burnout and clarify whether it is recognized in the DSM or ICD, or considered a distinct construct.

  2. Study Design: The methods section should clearly state that this is a narrative literature review, that no original data was collected, and that the aim is to synthesize existing findings, not to conduct new analysis.

  3. Inclusion/Exclusion Criteria: The exclusion of studies involving “narrow societal groups” needs clarification, please specify what is meant by this term.

  4. Structural Factors: While structural and sociocultural factors are mentioned in the introduction and abstract, they are largely absent from the sections on risk factors and treatment. Consider integrating them more consistently throughout.

  5. Figure 1: Figure 1 does not add meaningful content. Consider replacing it with a more informative visual, such as a model of contributing factors or the course of parental burnout.

  6. Organization: Several paragraphs, especially in the results and discussion, are too long. Breaking them into shorter, clearer sections would improve readability and flow.

  7. Tone and Precision: Be cautious of overly definitive or simplified statements. Aim for more balanced language that reflects the complexity of the topic.

  8. Overlap with Other Diagnoses: The manuscript would benefit from a brief discussion of how parental burnout overlaps with related conditions like depression or postpartum depression.

With revisions, this paper could make a useful contribution to the literature.

Author Response

(The authors gave the same response as above.)

Round 2

Reviewer 2 Report

Comments and Suggestions for Authors

I thank the authors for their effort to enhance the writing quality of the manuscript. 

Best of luck 

Author Response

Dear Reviewer,

We sincerely thank you for your assistance.

Best regards,  
The authors